# PCO-Based BLE Mesh Accelerator

**DOI:** 10.3390/s22145324

**Published:** 2022-07-16

**Authors:** Ivan Bukreyev, Hazal Yüksel, Ken Ho, Alyssa Apsel

**Affiliations:** 1Electrical and Computer Engineering, Cornell University, Ithaca, NY 14853, USA; kth62@cornell.edu (K.H.); aba25@cornell.edu (A.A.); 2Eridan Communications Inc., Mountain View, CA 94041, USA; hyuksel@eridan.io

**Keywords:** BLE mesh, duty-cycle, IoT, P2P, PCO, scalable synchronization

## Abstract

Bluetooth Low Energy (BLE) mesh networks enable diverse communication for the Internet of Things (IoT). However, existing BLE mesh implementations cannot simultaneously achieve low-power operation, symmetrical communication, and scalability. A major limitation of mesh networks is the inability of the BLE stack to handle network-scalable time synchronization. Pulse-coupled oscillators (PCOs) have been studied extensively and are able to achieve fast and reliable synchronization across a range of applications and network topologies. This paper presents a lightweight physical (PHY) layer accelerator to the BLE stack that enables scalable synchronization command with a PCO. The accelerator is a fully digital solution that can be synthesized with only the standard cells available in any silicon technology. This paper provides a detailed analysis of PCO-based BLE mesh networks and explores per-node system-level requirements. Finally, the analytical results are validated with measurements of a custom radio node based on the ubiquitous AD9364 transceiver.

## 1. Introduction

Internet of Things (IoT)-enabled devices have become ubiquitous. IBM estimates that the IoT market will grow to 75 billion devices by 2025 [1]. IoT has various definitions, but often refers to the low-power embedded devices that communicate with one another in an ad hoc fashion. This characteristic makes mesh networking desirable compared to the other network types (e.g., scatternets) because no single node is penalized and range and power requirements are relaxed. Because mesh network nodes connect dynamically and non-hierarchically, there is no fixed path between the source and the destination, so network coordination is required.

Bluetooth Low Energy (BLE) is a relatively new Bluetooth standard [2] designed to enable low-power communication with the ultimate goal of seamless connectivity and is a good candidate for mesh networking. According to the Bluetooth Special Interest Group (SIG), the number of Bluetooth^®^ devices is expected to surpass six billion by 2024. BLE is the fastest growing Bluetooth radio and consequently is becoming the new market standard for IoT [3]. The BLE mesh profile introduced in 2017 enabled a new paradigm for multi-hop networks by overlaying mesh networking on top of the standard scatternets and piconets [4]. There are now numerous applications for BLE mesh networks, such as the control of educational robots [5] and emergent ad hoc infrastructure [6]. However, the BLE mesh profile is based on flooding the three advertising channels and is fundamentally suboptimal for low-power operation since some nodes (e.g., relays) must have their receiver constantly powered [7]. Any other solution for BLE mesh must be implemented on top of the stack in the application layer, which necessarily incurs a communication overhead. Furthermore, in the literature there are few measured power results for Bluetooth multi-hop networks, as most results are based on analytical calculations or simulations [7,8].

The inability of existing solutions to deliver low-power mesh networking is directly related to the coordination problem—in order for the messages to flow from source to sink, the network needs to be synchronized, at least during the message transfer. Scalable synchronization is, at its core, a distributed communication approach. Figure 1 shows an example of distributed synchronization. Here, at some t=0, the first node (the leader) sends a low-power radio transmission to the nearby nodes. Upon receiving the message, the nodes in the network act as repeaters and re-broadcast this message to others. Eventually, the entire network receives the message and is synchronized.

There are a few important considerations in the approach shown in Figure 1. First, the sync message transmission and reception must be low-latency so that the maximum network time variation N × Δt is minimized. Second, the leader node is chosen arbitrarily based on some random frequency variation of the local clock (see Section 3.5). In other words, there are no special requirements on the leader node that would increase its power budget compared to others. Third, the synchronization protocol must be robust and infrequent to minimize power consumption of network maintenance.

Distributed synchronization has been observed in various species in nature. Certain firefly species are capable of achieving simultaneous flashing in the thousands, as shown in Figure 2a. During mating seasons, huge swarms of fireflies can flash to the same biological heartbeat or clock. This emergent behavior can be modeled mathematically with the pulsed-couple oscillators (PCOs). Mirrollo and Strogatz [9] first published the PCO synchronization model that has been since applied to a variety of engineering applications [10,11,12,13,14,15]. The conceptual diagram is shown in Figure 2b for three oscillators. PCO-based synchronization is of great interest to the designer because it creates a common reference frame for the local timing, which can enable aggressive duty-cycling of peer-to-peer (P2P) IoT nodes for low-power and long-range wireless communication.

This paper presents analytical and measured results for a proof-of-concept scalable and power-efficient solution to the BLE mesh networking problem via a slight augmentation of the physical (PHY) layer of the stack. A key design feature that determines the network’s ability to duty-cycle and scale is a symmetric synchronization protocol that does not rely on dedicated beacons or relays. Previous work has demonstrated low-power baseband timing circuitry that enables scalable synchronization for narrowband communication and rudimentary mesh networking with commercial radios [14,16]. This paper expands on the previous work with a system-level analysis of the PCO-based synchronization parameters, which is then validated by measured results. We note that techniques presented in this paper may be applied to a variety of wireless technology standards existing on the market (e.g., ZigBee and ANT+) and that we focus on BLE for evaluation purposes due to its ubiquity. The remainder of this paper is organized as follows. Section 2 addresses in detail the current BLE communication approaches and highlights the corresponding issues. Section 3 provides analytical models for the PCO-based network synchronization. Section 4 shows the measured results. We further note that the presented network synchronization techniques are not limited to the BLE standard.

## 2. Current BLE Mesh Solutions

BLE is the low-power variant of classic Bluetooth that was introduced in 2010. BLE operates in the same 2.4 GHz ISM band with forty 1 MHz wide channels spaced 2 MHz apart, as shown on the left of Figure 3. Out of the 40 channels, three are dedicated advertising channels. To avoid interference with other wireless communications in the ISM band (e.g., WiFi), BLE relies on an adaptive frequency hopping strategy by shifting communication to less-occupied or free channels. These devices require less power, have lower data rates, and support lower duty-cycle modes compared to their classic counterparts [17]. This section will address the structural problems that prevent BLE mesh networks from being low-power and how this can be solved with scalable synchronization.

### 2.1. Challenges with Existing BLE Mesh Implementations

Currently, there are many published BLE mesh protocols that fall into one of the two types of communication [7,8,18,19,20,21,22,23,24,25,26,27,28,29]. Table 1 lists some of the current BLE mesh solutions. Figure 4 shows the two common ways to organize communications in a mesh network: connection-oriented and flooding (connectionless). For connection-oriented BLE networks in Figure 4a, local piconets are loosely connected to form larger scatternets. A separate application layer protocol is required to manage scatternet formation/maintenance and multi-hop packet forwarding [7]. Bluetooth SIG has opted to use the flooding approach as the core underlying technique for its mesh standard. The BLE mesh profile [4] was adopted in 2017 and enabled the operation of connectionless multi-hop networks.

The flooding algorithm works roughly as follows. Each node acts as a relay and repeats the new incoming messages to all of the connected nodes. The process is repeated until the target nodes have received the message and the network is up to date. Controlled flooding algorithms limit the number of re-transmissions with techniques for caching and catching duplicate messages, and typically transmit an acknowledge message back to the source. Figure 4b shows the broadcasted message (red arrow) and its acknowledgement (blue arrow) propagating through the network. The BLE Mesh profile works on a publish/subscribe model where individual nodes can listen for and transmit to specific network addresses. In this model, both unicast (point-to-point) and group addresses are supported. However, each node must have its receiver enabled the entire time in order to catch any possible messages, or risk missing communication.

While BLE mesh offers great promise on mobile platforms, there are few published works with measured power analysis [7,8,30,31]. Even though the mesh protocol is based on BLE, the power consumption is relatively high due to the way mesh networking is handled [30,32]. To better understand the trade-offs of connected vs. connectionless BLE mesh network solutions, consider two representative literature implementations: connection-oriented FruityMesh [19] and flooding Trickle [18]. In their 2017 paper (before BLE Mesh profile was released), Murillo et al. evaluate the power consumption and packet delivery ratio of these two approaches [33]. They use a realistic mesh network topology as shown in Figure 5. They use Nordic Semiconductor nRF52 BLE development boards with transmitter (TX) power set to −20 dBm for node separation of 1.5 m. The power consumption for a FruityMesh node is 9.91 mW (the reported number is 9.4 mW, which is inconsistent with the number obtained using their Equation (Equation 3)), of which about 80% is used for protocol maintenance to exchange keep-alive messages with the neighbors. For the flooding counterpart Trickle, the reported number is 28.5 mW, of which about 58% is spent on keeping the receiver (RX) powered continuously. This study highlights how the power consumption of the existing BLE mesh network protocols is relatively high compared to the regular BLE connections [33].

Symmetrical communication and low-power operation is a challenging combination to achieve simultaneously. In the example above, Trickle protocol consumes 16.8 mW to keep the RX enabled continuously and FruityMesh consumes the majority of the power on network maintenance [33]. Both Trickle and FruityMesh modify the application layer of the stack, thus forgoing the potential benefits of a PHY layer implementation. It is clear from the above example that both flooding and connection-based implementations need to solve a coordination problem in order to successfully pass down messages from source to sink.

### 2.2. Time Synchronization and Duty-Cycling in Mesh Networks

One of the challenges for BLE mesh is that due to the lack of a scalable synchronization strategy, the nodes are not low power, as RX must have a high duty-cycle. Low-power operation can be achieved only by duty-cycling the radio during inactive periods. However, for any communication to occur, RX must be enabled precisely when TX is broadcasting. Synchronous and asynchronous duty-cycling strategies are illustrated in Figure 6. For an asynchronous protocol such as the BLE mesh profile, a node that has a message to transmit starts sending a preamble at a regular interval. At some point, the receiver wakes up and sends a response packet, which prompts the transmitter to deliver the message. The advantage of this strategy is its simplicity because no other prearranged schema is required. The obvious disadvantage is that neither RX nor TX are aware of each other’s timing. For example, the BLE mesh profile requires a 100% duty-cycle to scan the advertisement channels for incoming messages. This further turns into a guessing game because the message can be on any of the three advertising channels. Time synchronization is required to communicate efficiently.

For the synchronous duty-cycling approach shown in Figure 6, the two nodes decide on the communication window ahead of time and turn on the radio for the minimum required time. Since any two radio nodes will have different clocks, their local time will constantly drift, as described in Section 3.4. Therefore, synchronous duty-cycling requires a mechanism for synchronizing clocks. While a number of synchronization protocols are available (e.g., RBS, TPSN, or FTSP), they rely on exchanging timestamped packets and are consequently costly to scale [32,34,35]. BLE presents additional challenges because the application layer does not have access to the timestamps of the incoming messages in the PHY layer.

**Figure 6 sensors-22-05324-f006:**
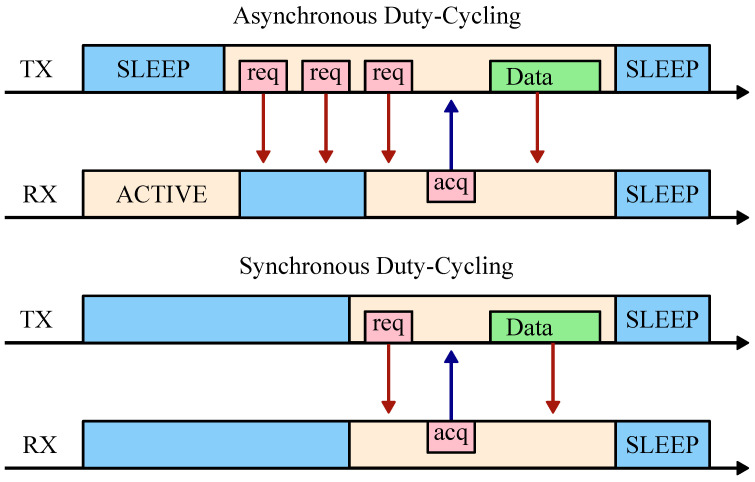
Illustration of synchronous vs. asynchronous duty-cycling. In the asynchronous case, request packets (req) must be sent continuously by the sender until the receiver responds with an acknowledge packet (ack) before communication can take place. In the synchronous case, the active and sleep regions are aligned based on a synchronization scheme.

There are several solutions to the BLE synchronization problem in the literature. In [36], the authors achieve a synchronization accuracy of 37.78 ms by using a dedicated synchronization center microcontroller. In [37], the authors report a synchronization accuracy of 14.19 μs by using the establishment of a BLE connection event as the basis. In [38], the authors report a synchronization accuracy of 750 μs by forcing two BLE nodes to establish a connection at a fixed interval and using the time difference between connect events for estimation. In [39], the authors achieve a synchronization accuracy of 9–17 μs with a software interrupt technique and 0.9 μs with the addition of power measurement circuitry. In [5], the authors introduce a protocol to synchronize task execution in a multi-hop scenario and achieve task synchronization on the order of tens of milliseconds. Without hardware modification, the best reported synchronization accuracy is in the order of tens of microseconds. Notably, the authors in [40] introduce BlueSync, a solution that achieves excellent synchronization quality between a couple nodes, but notably it has master-minion synchronization asymmetry and is not intended for multi-hop networks. We will address this scheme as an orthogonal strategy to the approach presented in this paper in Section 4.5.

Despite the apparent simplicity and the lack of duty-cycling of BLE mesh, further scaling is also nontrivial due to the master-minion asymmetry. Each node repeats incoming new messages so that they are relayed further in the network. For a large enough network, a new message may propagate through many routes and cause a broadcast storm, a network condition that represents an increase in collision probability, network congestion, and packet loss rate [32,41]. In connection-oriented BLE mesh, the number of minions connected to a master is limited due to the need for the master node to reset its radio frequency per each minion. Furthermore, each pair of connected nodes must keep unique timing, which becomes a challenge over multiple network hops [17].

### 2.3. Proposed PCO-Based BLE Mesh Accelerator

Synchronization protocol selection is a major design consideration when determining the network’s ability to duty-cycle in BLE mesh. Several literature sources noted that a combination of flooding and connection-oriented protocols may achieve optimal performance [33,42]. However, most proposed solutions are theoretical, and out of those with measured results, all deal with the higher levels of abstraction of the BLE stack.

We present a circuit-level solution to the mesh networking challenges. We propose a slight PHY-layer augmentation of the BLE stack utilizing PCO-based synchronization, as shown in Figure 7. Notice how the communication scheme is similar to the standard synchronous duty-cycling in Figure 6 with the important difference that global synchronization is carried out via distributed action of every node. The local PCO phase represents consistent time for all the network nodes, such that a predetermined communication time slot can be agreed upon ahead of time. The modification is a digital logic-only application-specific integrated circuit (ASIC) that has a low area profile and interfaces directly to the analog to digital converter (ADC) of the commercial radio front-end. The accelerator enables a global synchronization command, the syncword, with the correlator and the PCO circuitry. The PCO syncword acts as a distributed and periodic synchronization broadcast that allows the nodes on the network to communicate efficiently without the need to back-calculate clock differences or exchange timestamps. Our experimental results validate the analytical calculations and show that PCO-based synchronization facilitates the establishment of low-power (duty-cycled) P2P mesh networks.

## 3. Synchronization Analysis and Requirements

The previous section discussed challenges with existing BLE mesh networks and addressed the need to solve a coordination problem in order to efficiently pass messages through multiple hops of the network. A scalable PCO-based synchronization accelerator for the BLE PHY layer was proposed. This section presents network synchronization analysis and further refines the requirements for mesh networking as follows. In Section 3.1 we introduce the concept of PCO synchronization in more detail and present simulation results. In Section 3.2 we define our test platform and justify the choice of radio. In Section 3.3 we address radio power usage vs. PCO period and coupling sequence duration. In Section 3.4 we address the effects of the crystal quality on the minimum system power. In Section 3.5 we summarize the results.

### 3.1. PCO-Based Synchronization

Massive-scale mutual synchronization occurs naturally among fireflies, with some species capable of achieving synchronous flashing in the thousands. This has been studied extensively in the literature and can be modeled by treating each insect as a pulse-coupled oscillator (PCO) [9]. The oscillator’s phase state function ϕ(t) is reset to zero upon reaching a threshold (ϕ(t)=1), at which point the firefly sends out a coupling pulse (a flash of light) that advances nearby fireflies’ phase by some fixed amount. For synchronization, ϕ(t) must be monotonically increasing ($\phi^{\prime }\left(t\right)>0$) and concave down ($\phi^{"}\left(t\right)<0$), such that depending on the exact timing of the incoming pulse, the remaining phase is reduced by a varying amount. It was shown in [9] that for a smooth and continuous state function, the oscillators will necessarily synchronize. In [43] it was numerically demonstrated that the synchronization will happen even if the state function is digital and discrete.

PCOs have been used in various applications, e.g., as a mechanism for packet routing for chip-satellites [15]. In this work, we utilize PCOs as the driving force for mesh network synchronization. We modified an event-driven simulator based on the one originally developed in [10] to be compatible with narrowband communications to study PCO-based synchronization of BLE devices. For example, consider a network of 50 radio nodes randomly scattered on a 100 by 100 meter grid with a 1 s PCO period. Figure 8a shows the randomly generated network topology. Figure 8b shows phase offsets of all 50 nodes relative to the node with the fastest natural period, i.e., the leader node. Figure 8 shows that within 15 leader cycles (≈15 s), the network reaches a stable configuration with four distinct phase bands. These bands correspond to the number of hops away from the leader node, similar to what is depicted in Figure 1. As the leader fires, it resets nodes in its vicinity, which then reset the following nodes, until in three hops the entire network is reset. Figure 8b also represents the latency of the coupling signal, i.e., the syncword, propagation through the network when it is locked to a leader. Notice that 38 μs (see Section 3.4) is the maximum timing difference between any two nodes that are in range of one another. Therefore, the network synchronization latency is 38 μs, which is several orders of magnitude faster than what is achieved in prior art [38,39].

### 3.2. System Configuration

To evaluate synchronization and duty-cycling performance of a BLE system with our ASIC accelerator, we found that the optimal solution is to utilize a single-chip software-defined transceiver that has dedicated control of its key components. For this work, we chose to use the AD9364 commercial radio front-end due to its extreme flexibility. A popular choice for BLE evaluation is the Nordic Semiconductor nRF series SoCs, but the onboard radio does not have dedicated connections for integration with the PCO circuitry. The AD9364 can interface directly with the ASIC accelerator via the digital I/Q baseband and its precise timing control allows for quick wake-up and sleep cycles, and consequently, accurate duty-cycle measurements.

Figure 9 shows the functional top-level block diagram of the implemented P2P radio node that is based on prior work in [16]. Each node comprises a custom RF (radio frequency) board with an AD9364 transceiver (the radio), a crystal oscillator and miscellaneous support circuitry, the ASIC accelerator chip, and a ZedBoard. The radio is used for data and syncword transmission. The digital data interface connects the chip with the radio via ZedBoard level shifters. The ZedBoard’s integrated logic analyzer enables precise radio state transition measurements. The ASIC accelerator chip processes digitized radio data and decodes and detects a synchronization packet, the syncword. Whenever the node’s PCO reaches its threshold, the accelerator generates an encoded syncword and initiates the transmission process.

For narrowband communications, the communication of the syncword sequence is not instantaneous. In [44] it was shown that a 63-bit Kasami sequence (transmitted as a 64-bit encoded) is a good trade-off between the bit error rate (BER), implementation complexity, and length. Fast detection of the syncword is essential for low-power operation (more details in Section 3.4), and given that the network starts up in a random state, the detection must be asynchronous. The key block of the ASIC accelerator is the signal processing core that performs an asynchronous detection of the digitized radio signal against the expected sequence. The core is designed to detect binary phase-shift keying (BPSK)-encoded signals with low latency by utilizing a differential detector. The signal processing core is able to detect the syncword regardless of the phase and frequency mismatch between the TX and the RX local oscillators (LOs). Prior works describe in detail the operations performed by the signal processing core [14,16]. If the syncword is detected, the state of the PCO is advanced. Whenever the PCO reaches its threshold, either from external coupling or a self-reset, the sequence generator wakes up the radio and initiates the syncword transmission.

### 3.3. Radio Power Analysis

For long-range mesh networks, a high-power, narrowband transceiver is required. Therefore, the length of the syncword and its repetition frequency will predominantly dictate the average system power. Furthermore, for successful syncword detection, the receiver must be enabled sometime before the PCO self-reset to account for the frequency variation (detailed discussion in Section 3.4). We analyze the power overhead vs. syncword length (or TX data packet in the generic case), RX on-window, and repetition frequency based on the AD9364 transceiver datasheet. Figure 10a shows that for sufficiently large $T_{PCO}$ (repetition frequency), the radio power approaches the leakage values. Figure 10b shows power consumption vs. radio on-time for a 1 s repetition frequency. For both plots, the RX window is set to be twice as long as the TX window.

There are two critical observations. First, for fixed TX and RX windows, the radio power asymptotically approaches the leakage value of 450 μW (Figure 10a) with less frequent communication. On the other extreme, power grows linearly towards fully-on value for more frequent communication. Therefore, there is an optimal trade-off between power consumption and the syncword broadcast rate. Figure 10a shows that $T_{PCO}>0.5\;s$ gives linearly diminishing returns in power reduction. Second, for a fixed $T_{PCO}$, there is some range of packet lengths that consume negligible power compared to the leakage. This means that in addition to a syncword transmission of 64 bits, up to 1000 bits of data incur virtually no power increase from the leakage value of the radio. Both plots shows the effectiveness of duty-cycling for reducing the average power.

To incorporate duty-cycling, the radio circuitry should interface with a duty-cycle controller that can keep track of the synchronization state and safely turn off-and-on various components according to a schedule. The operation of the controller is described in more detail in prior work [16]. Figure 11 shows the state transition timing for the AD9364 radio when receiving and transmitting the syncword. In a nutshell, the RX wakes up sometime before the PCO natural reset and listens for a potential syncword from the other nodes. Then, whenever the PCO counter reaches its threshold, the TX broadcasts a syncword and the radio is brought back to the sleep state. There are several optimizations that can be performed. For example, the RX can be disabled as soon as an incoming syncword is detected and while the TX is still broadcasting. The next section will address system latency and the minimum achievable duty cycle and will further elaborate on the timing diagram in Figure 11.

### 3.4. System Parameter Analysis for Mesh

For narrowband communication, the transmission and reception of the syncword take time proportional to the sequence length. For the PCO-based network to function properly, every node must receive and transmit the syncword. The total time overhead of syncword communication consists of RX syncword reception, the speed of light propagation delay, signal processing latency, and TX syncword generation. The cumulative delay $t_{delay}$ sets the fundamental duty cycle limit and is given by
(1)$$D_{min}^{\prime }=\frac{\left(t_{syncword}+t_{core}+t_{latency,RX}+t_{latency,TX}\right)+t_{flight}}{T_{PCO}}=\frac{t_{latency}+t_{flight}}{T_{PCO}}=\frac{t_{delay}}{T_{PCO}}\;.$$

Figure 12 illustrates where in the signal path each delay contribution occurs. Specifically, $t_{syncword}$ is the duration of the coupling sequence, $t_{core}$ is the ASIC accelerator processing latency, $t_{latency,RX}$ and $t_{latency,TX}$ are the receiver and transmitter path latencies that are usually dominated by the finite impulse response (FIR) filters, and $t_{flight}$ is the propagation delay. For a BLE system, $t_{flight}$ is negligible because it adds only 0.33 μs of delay for a 100 m node separation, which is shorter than the time duration of a single BLE symbol at the maximum data rate. Therefore, for the purposes of this analysis we can treat $t_{latency}\approx t_{delay}$.

However, in a real system there will be further duty-cycling degradation due to the non-idealities of the reference crystal. The two important parameters are frequency stability and jitter, and both have direct implications on the minimum RX window size and synchronization quality. Figure 13 visualizes the two non-idealities. Frequency stability results in a uniform distribution of the center frequency for every individual oscillator and is usually expressed in parts per million (ppm) of the nominal frequency. Then, due to various noise sources, the oscillation spectrum is spread around the center frequency as a phase noise skirt. Phase noise in the frequency domain is equivalent to RMS period jitter in the time domain.

Using an event-driven simulator based on the one originally developed in [10], we can quantify $D_{min}^{\prime }$ vs. the oscillator ppm. We use the simulator described in Section 3.1 with maximum coupling. The network consists of 50 nodes randomly distributed on a 100 m by 100 m grid with a coupling range of 30 m. Each node uses a 19.2 MHz crystal oscillator with jitter set to zero. The 64-bit syncword is transmitted at 2 MSPS and $t_{delay}$ is set to 38 μs, which includes all of the latencies and processing delays with margin. The results are shown in Figure 14a. For the perfect crystal, the $D_{min}^{\prime }$ approaches the theoretical value of 0.0038 % that is set by $t_{delay}$, and increases linearly with an increasing oscillator frequency mismatch.

The effect of jitter on $D_{min}^{\prime }$ is not immediately apparent because it predominately affects the synchronization quality. For example, for a 19.2 MHz crystal with the RMS period jitter of 25 ps and $T_{PCO}$ of 1 s, the accumulated RMS $T_{PCO}$ jitter is only 110 ns ($\sigma_{T_{PCO}}=\sqrt{N}\times \sigma_{T_{ref}}$), much shorter then the $t_{delay}$. In [44] it is explained that a high jitter value may cause the second-fastest node to temporarily become the leader. In this case, the real leader is reset by a slower node, which causes the network to temporarily lose lock. The collective effect of both clock non-idealities can be written as
(2)$$t_{crystal}=2*\left(ppmT_{PCO}+3\sigma_{T_{PCO}}\right)\;,$$
where a factor of three accounts for 99.7% variation of PCO jitter and a factor of two takes care of both minimum and maximum variation away from the nominal. The collective degradation of $D_{min}^{\prime }$ based on both clock non-idealities can be then written as
(3)$$D_{min}=\frac{t_{delay}+t_{crystal}}{T_{PCO}}\;.$$

Effectively, Equation (Equation 3) is the same as Equation (Equation 1), but with the effects of crystal non-idealities included. Figure 14b shows the effect of 50 ppm crystal with RMS period jitter of 25 ps on the RX window overhead from Figure 11. The log–log scale shows that accumulated error due to the crystal is 10.2 μs during the first 100 ms since timing errors are reset by the PCO, which is negligible compared to $t_{delay}$ of 38 μs. However, at 1 s the accumulated error is 100.7 μs and above that, Figure 14b shows that crystal non-idealities dominate all of the RX overhead and set the limit on the minimum duty-cycle.

Armed with Equation (Equation 3), we can assess the cumulative effect of all non-idealities on $D_{min}$. For any node, the radio operation procedure is as follows. First, the radio wakes up and enters the receive state, indicated by RX On in Figure 11. This time is dependent on the particular radio implementation, and in the case of our system with AD9364, it also includes the warm-up time of the receive ADC and FIR filter latency. After $t_{syncword}$, the TX is powered and ready to transmit the syncword in case the node is reset by the incoming coupling, as indicated by TX On. Whenever the node resets at $t=T_{PCO}$, the RX is disabled and TX starts transmitting the syncword. Note that the RX window is stretched by the accumulated error captured by $t_{crystal}$.

### 3.5. Analysis Summary

The goal of Section 3 was to demonstrate the feasibility of the PCO-based BLE mesh networking. This section demonstrated several key takeaways for a PCO-based BLE mesh network. First, we showed in simulation that a four-hop PCO network of 50 nodes with a BLE-compatible data rate and range will synchronize and then maintain synchronization with the low latency of 38 μs. Second, we showed that a PCO syncword exchange rate of $T_{PCO}>0.5\;s$ will result in a minimal average system power increase (Figure 10a). Furthermore, we showed that a node can transmit an additional 1000 bits of data over the 64-bit syncword without deviating significantly from the leakage power (Figure 10b). Third, we quantified the effects of crystal non-idealities on the minimum achievable duty-cycle (Figure 14a). We showed that for $T_{PCO}<100\;\mathrm{ms}$, $D_{min}$ is dominated by $t_{delay}$ (Figure 14b). Furthermore, for a BLE system, the effects of $\sigma_{T_{ref}}$ are negligible compared to the ppm of the crystal. Collectively, the findings highlight the feasibility of PCO-based BLE mesh networking to remain below a 0.02 % duty-cycle (Figure 14a).

There are additional considerations related to the synchronization quality and robustness that are studied in great detail in other works [10,13,43] but not explicitly analyzed here. We will provide a qualitative summary of their findings. Network synchronization relies on natural frequency variation between different radio nodes such that the fastest node, the leader, will be the first to reset and initiate the reset for the rest of the network (see Figure 1 and Figure 8). The node’s radio is first used in RX mode to listen for possible transmission from faster nodes, and then in transmit mode when the node reaches reset. Sections III D and E in [16] provide more detail.

However, for the network to maintain synchronization, the leader node must be faster than the second-fastest node to prevent temporary leader switching. The period difference should be large enough to compensate for the jitter of the second-fastest node. Effectively, there is a natural trade-off between synchronization quality and power savings based on the reference crystal—a lower ppm crystal results in a lower $D_{min}$, but reduces the likelihood of a single fastest node that can lead the network, thereby degrading stability. Table 2 summarizes the system-level trade-offs discussed in this section.

## 4. Measured Results

To validate the analytical results with our PCO-based ASIC accelerator, we implemented a custom radio node based on the AD-FMCOMMS4-EBZ evaluation board with the ubiquitous AD9364 transceiver. First, we characterized each node individually to verify its performance and build the complete power profile. Second, we performed synchronization measurements to verify a subset of the theoretical results. Finally, we verified a representative mesh configuration to demonstrate a synchronized multi-hop data transmission.

### 4.1. Implemented System

The implemented system is shown in Figure 15 and comprises a ZedBoard, a custom RF board, and a breakout board for the ASIC accelerator. The ASIC accelerator is fabricated in a 180 nm six-layer CMOS process with only five digital routing layers. The active area of the chip is 1.14 mm by 1.34 mm with 26.8% density. In this design, the chip area was pad-limited and could be scaled down to a significantly smaller footprint (below 0.4 mm^2^), especially in a more advanced digital process.

We used a 19.2 MHz crystal that is common in 3G applications with an accuracy of 50 ppm to run the PCO portion of the ASIC accelerator. The AD9364 TX and RX ports are connected via a 2450BL15B050 balun to an ADG918 RF switch that collectively incur approximately 2 dB insertion loss. AD9364 supports vastly higher data rates than is needed for BLE, so we configured it to transmit and receive at 2 MSPS, corresponding to the standard’s maximum data rate.

The chip processes digitized radio data and decodes and detects the syncword with approximately two baseband clock period latency ($t_{core}$). The chip operates in two different clock domains. The PCO and duty-cycle controller are running constantly at the crystal clock, while the signal processing core and sequence generator can be clock-gated and run at the baseband data clock.

**Figure 15 sensors-22-05324-f015:**
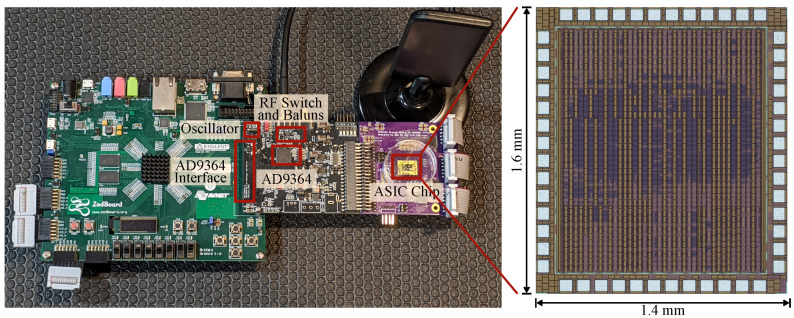
The implemented radio node consists of a ZedBoard, a custom RF PCB with AD9364, and an ASIC accelerator board, and is connected as shown in Figure 9. The ASIC chip is pad limited with 26.8% standard cell density in a 180 nm six-layer CMOS process.

### 4.2. Single Node Performance and Power Measurements

First, we have verified the functionality of a single node by measuring the syncword detection error rate as detected by the ASIC chip. The measurement setup is shown in Figure 16a and consists of a free-running node with duty-cycling disabled, which is locked to an RF signal generator running with a faster PCO period. Figure 16b shows the syncword transmission error rate vs. radio power for the implemented node. In order to achieve the lowest $D_{min}$, it is prudent to reduce radio on-duration as much as possible. Figure 11 shows the state transition timing for the AD9364 radio and the transition sequence that we have implemented for this study. Even though BLE is operating in TDD mode, AD9364’s FDD mode allows for a more flexible timing for the transmission and reception of the syncword.

The AD9364 radio is highly agile, but is significantly over-provisioned for BLE applications. As such, the system power can swing three orders of magnitude in a few microseconds, making it difficult to measure. We developed a steady-state power profile for every AD9364 state shown in Figure 11. By measuring transition times between the different states, we reconstructed a realistic power profile for data detection and transmission of the system. Table 3 shows the power breakdown of both fully-on and duty-cycled operation. The duty-cycled system power stays below 1 mW.

### 4.3. Synchronization Measurements

The time it takes the network to reach a synchronized state takes on the order of tens of PCO cycles (see Figure 8b) and depends on the exact node distribution and the configured PCO parameters [43]. After the network synchronizes, all circuitry except for the PCO, duty-cycle controller, and the clock source can be duty-cycled to conserve power (see Table 3). The measurement in Figure 17 shows a scope capture of the syncword detection and transmission procedure for a node in the network. The data rate for the syncword is configured for 2 MSPS, which results in a 32 μs-long sequence ($t_{syncword}$). Shortly before the PCO counter resets, the duty-cycle controller initiates the radio wake-up sequence (orange line in Figure 17). From the initial wake-up, the entire process takes 204 μs to transition back to the SLEEP state. When clocks are disabled, the radio power consumption is dominated by the leakage. The node in Figure 17 transmits the syncword when its PCO counter reaches the threshold. However, the RX is enabled and functional for an additional 132.7 μs ($t_{crystal}+t_{syncword}$) before the TX to detect the incoming syncword from other nodes within the oscillator ppm tolerance window (note that $t_{latency,RX}$ is folded into the total active radio time).

**Figure 17 sensors-22-05324-f017:**
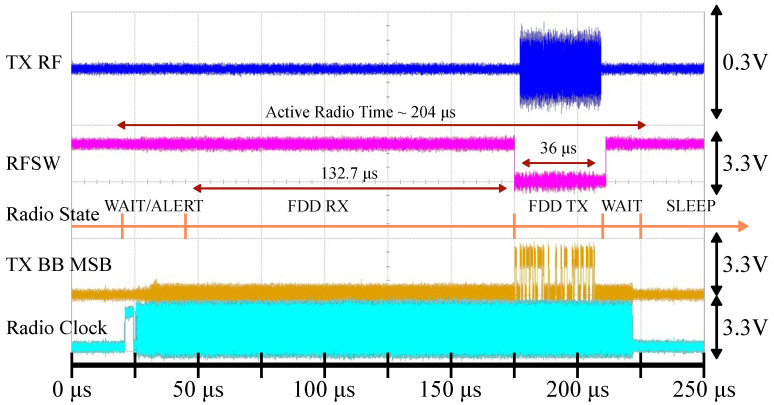
Duty-cycled syncword detection demonstration for $T_{PCO}=1\;s$. The duty-cycle controller first brings the radio to the FDD state based on the PCO counter value. After the syncword is transmitted, the radio returns to the SLEEP state. The RX window is opened for 132.7 μs according to Equation (Equation 3), the TX window is 36 μs ($t_{syncword}+t_{latency,TX}$, with some margin), and the active radio time is 204 μs. Signals from top to bottom: TX RF output at the antenna (blue), RF switch direction where logic high selects RX (purple), TX baseband data (MSB) output of the chip (yellow), AD9364 clock (teal).

In addition to the syncword, each node in the network needs to transmit and receive actual data. Each syncword detection event resets all timing errors such that a reliable communication window can be established at the agreed-upon time. This process is illustrated in Figure 18 for a 1 s PCO period. In this demonstration, a node is configured to open an RX communication window for 33 μs ($t_{crystal}+t_{syncword}$, rounded up), 200 μs after syncword transmission, before returning to the sleep state. The amount of RX overhead depends on how much time has elapsed since the last PCO reset and increases as shown in Figure 14b.

To quantify the limits of duty-cycling, let us consider what would happen if the FDD RX duration is reduced from what it is in Figure 17. For this measurement, we set up two nodes with a 200 μs PCO period, connected by a −70 dBm link, free of any interference or obstructions, as shown in Figure 19. The period is deliberately shortened for the syncword error rate measurement. Theoretical calculations in Section 3.4 predict that the maximum RX window size (rounded up) for a 200 μs PCO period is 33 μs (Figure 14b), though the exact number depends on the particular frequency deviation between the two nodes (Figure 13).

The results are shown in Figure 20a. The error rate is better than $10^{-6}$ for an RX window larger than 16 μs, after which point it rapidly increases since the few initial bits of the syncword are cut off and the detection fails. The observed $D_{min}$ of 0.0055% (based on the total active radio time) for this measurement is overlaid against the theoretical data and is shown in Figure 20b.

**Figure 20 sensors-22-05324-f020:**
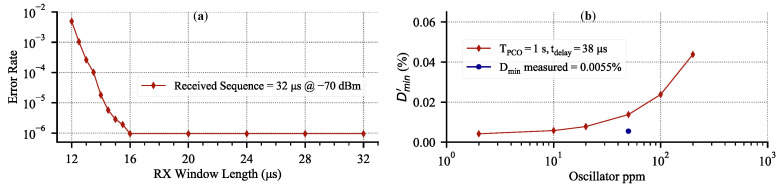
Error rate measurement for a two-node system at a fixed separation. (**a**) Syncword detection error rate vs. progressively reducing RX window for $T_{PCO}=1$ s. This is equivalent to moving **RX On** to the right until failure in Figure 11. The plot shows that for an insufficient RX window, the error rate rapidly increases. (**b**) The corresponding measured data overlay for Figure 14a.

### 4.4. Multi-Hop Measurements

For any network-level communication, a complete communication protocol (e.g., those listed in Table 1) is required. While it is possible to implement any of the published protocols on the ZedBoard, we opted to reuse the existing syncword detection and generation hardware to demonstrate basic multi-hop data transfer. In our protocol, messages are encoded as special sequences that are detectable only by nodes that choose to look for them. In this fashion, after syncword detection/transmission, a node reprograms its correlator for a different sequence to transmit and/or receive as data. This basic communication approach is suitable to demonstrate duty-cycled multi-hop data transfer with minimal additional hardware and software by using our PCO-based accelerator.

A measurement setup for the duty-cycled three-node multi-hop data transmission with our rudimentary protocol is shown in Figure 21. We set up node A (source) and node C (sink) to be out of radio range of each other while both are in range of node B. Therefore, the source at node A needs to hop through node B in order to reach the sink at node C. Each node is configured to wake up twice per a 1 s PCO cycle, identical to the measurement in Figure 18. During the second wake-up event, node A is configured to transmit a code designated for node B (TXA). Similarly, node B is configured to listen for a transmission from node A (RXB). If, and only if, node B successfully detects node A’s data packet, it will broadcast a message for node C (TXB) on the following PCO cycle. Finally, node C is configured to listen for node B’s message (RXC). For visual clarity during testing, node C is configured to toggle an LED (TXC) on its ZedBoard if, and only if, it successfully detects a data packet from node B, thereby demonstrating multi-hop connectivity.

For the measurement in Figure 21, we are interested in the minimum duty-cycle and power consumption of each of the three nodes. The baseline power consumption is given by Table 3 and is 982 μW for all three nodes. The second communication window is chosen to be 200 μs after the PCO is reset in order to minimize $t_{crystal}$. Effectively, communicating shortly after the PCO reset allows for the shortest RX and TX windows (Figure 14b). The additional 32 μs communication window consists of receiving or transmitting an 8-byte data payload (an encoded sequence) at a 2 MSPS data rate, which adds 70.66 μs or 85.73 μs to the active radio time, respectively. In total, this adds an additional 20.5 μW for node A (TX only), 16.8 μW for node B (alternating RX and TX), and 13.1 μW for node C (RX only).

### 4.5. Discussion and Conclusion

It is difficult to directly compare our PCO-based mesh accelerator with other literature solutions. We demonstrate a means to achieve scalable synchronization that can be layered with many different communication protocols. However, due to the distributed nature of PCO synchronization, our technique automatically forms an emergent network topology with automatic hop formation (see Figure 8). Compared to other synchronization schemes [5,36,37,38,39], we achieve significantly higher synchronization accuracy at the cost of additional hardware. The presented accelerator is lightweight, but needs to be integrated into the PHY layer (radio ADC), which likely implies a new hardware design. As described in Section 3.2, many standard Bluetooth solutions are highly optimized and do not allow for easy integration with the radio front end. In the near future, our solution can be integrated with custom evaluation platforms based on any software-defined radios, such as the AD936x series.

To the best of our knowledge, our approach provides the highest network synchronization accuracy and the lowest achievable duty-cycle compared to any existing BLE mesh solution. BlueSync [40] reports synchronization errors as low as 320 ns between synchronization master and minion. BlueSync is based on three underlying approaches: timestamping, frequency–drift estimation, and ticks adjustment. Effectively, their approach is to use timestamps to accurately calculate clock offset and drift between the two synchronizing nodes with the intention to update timing infrequently. In our scheme, timing errors accumulate over time, as shown in Figure 14, which is a direct comparison to the offset-only lines of Figure 6 in [40]. The approach in [40] can be overlaid on our PCO-based mesh network to further increase synchronization accuracy.

We proposed a lightweight PCO-based accelerator to augment the BLE physical layer with a scalable synchronization command to enable multi-hop mesh networking. We presented the analysis to show how synchronized BLE nodes can be aggressively duty-cycled to save power even with an overprovisioned AD9364 transceiver. We provided simulation results that corroborate analytical estimations. Finally, we verified individual node power consumption against estimates and provided a rudimentary mesh networking data transfer.

## Figures and Tables

**Figure 1 sensors-22-05324-f001:**
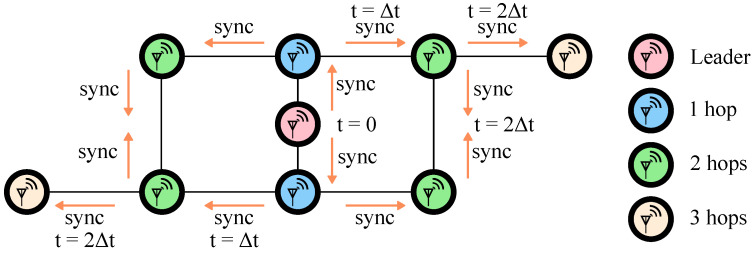
Synchronization packet (sync) propagating through the network starting from the leader node. Colors highlight hops away from the leader. In this work, a 64-bit binary phase-shift keying (BPSK)-encoded Kasami sequence is used.

**Figure 2 sensors-22-05324-f002:**
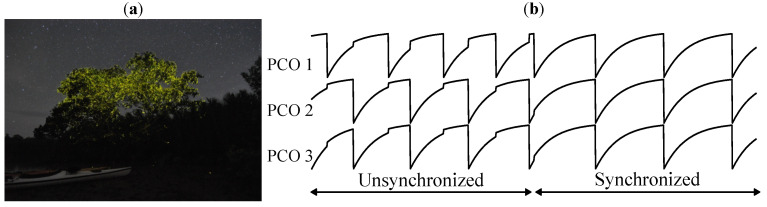
(**a**) A still shot that shows how fireflies can synchronously flash. (**b**) PCO synchronization demonstration. Diagram shows how three PCOs with random initial phase eventually synchronize.

**Figure 3 sensors-22-05324-f003:**
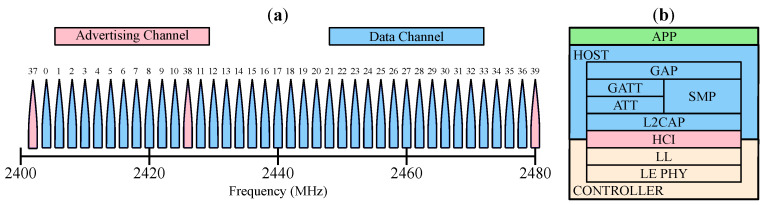
(**a**) BLE channel spacing. (**b**) BLE stack.

**Figure 4 sensors-22-05324-f004:**
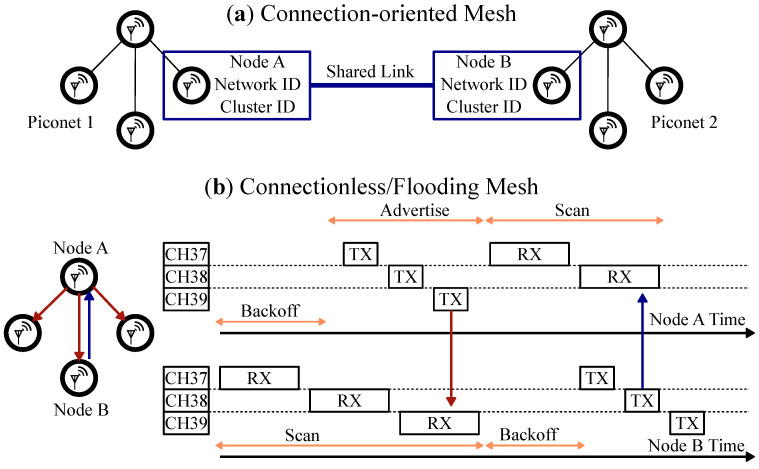
Conceptual representation of the two types of BLE mesh topologies. (**a**) Connection-oriented mesh network. (**b**) Connectionless or flooding mesh network.

**Figure 5 sensors-22-05324-f005:**
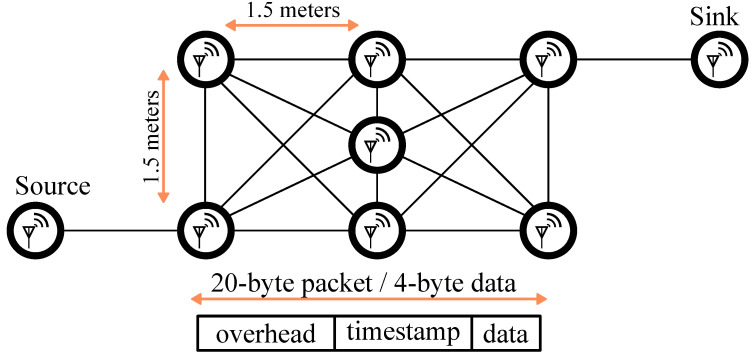
Power consumption benchmark of connection-oriented FruityMesh vs. flooding-based Trickle BLE nodes in a network routing scenario from [33].

**Figure 7 sensors-22-05324-f007:**
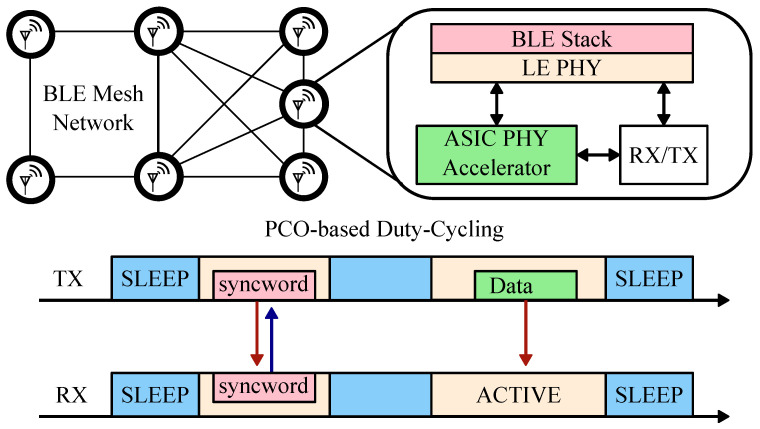
Proposed stack modification for scalable BLE mesh.

**Figure 8 sensors-22-05324-f008:**
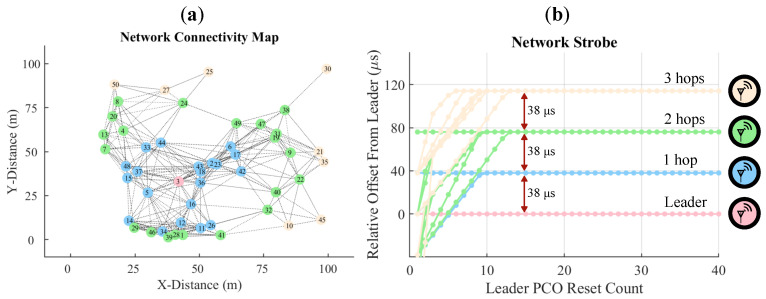
(**a**) Network of 50 nodes generated by a uniform distribution with a 30 m effective range on a 100 by 100 m grid. (**b**) PCO reset offset of the 50 nodes relative to the fastest (leader) node across 40 leader cycles. Synchronization is achieved within the first 15 cycles, after which maximum synchronization latency is 38 μs between successive hops.

**Figure 9 sensors-22-05324-f009:**
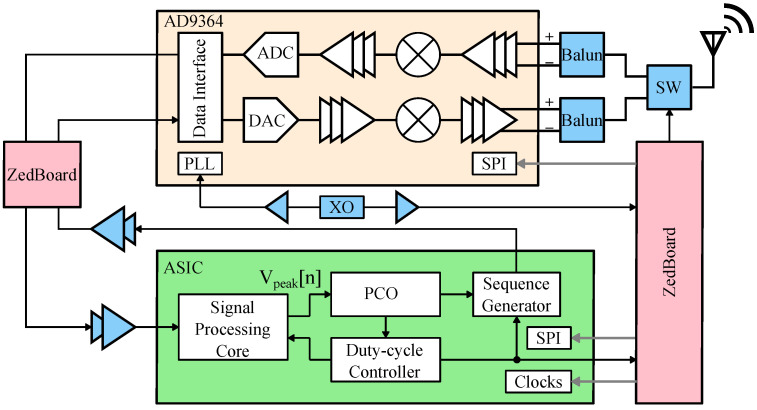
A functional block diagram of the peer-to-peer (P2P) radio node. AD9364 transceiver’s digital IO is routed to the ASIC chip for syncword detection and generation via PCB level-shifters. A shared crystal provides clock for the entire system. A ZedBoard is used to initialize the system and perform miscellaneous functions.

**Figure 10 sensors-22-05324-f010:**
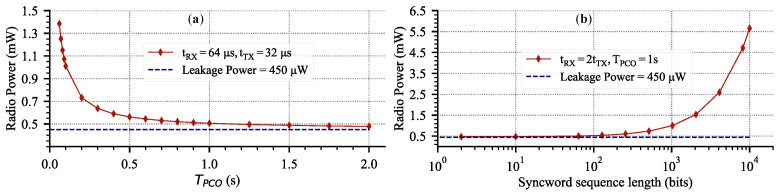
Estimated AD9364 radio power using datasheet values for a 2 MSPS data rate, 10 MHz bandwidth at 2.4 GHz, 7 dBm output power. (**a**) Power vs. PCO period for a 64-bit syncword sequence. (**b**) Power vs. syncword length for a fixed PCO period.

**Figure 11 sensors-22-05324-f011:**
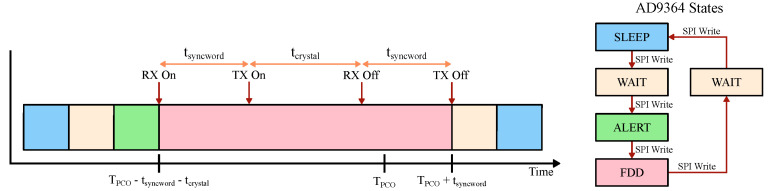
Optimized state transition diagram of the radio. TX can lag RX by $t_{syncword}$. RX must be enabled before the natural PCO reset to account for the reference crystal uncertainty. Reference crystal quality and syncword duration set the limit on how the system components can be duty-cycled. RX and TX latencies are omitted from the diagram for visual clarity.

**Figure 12 sensors-22-05324-f012:**
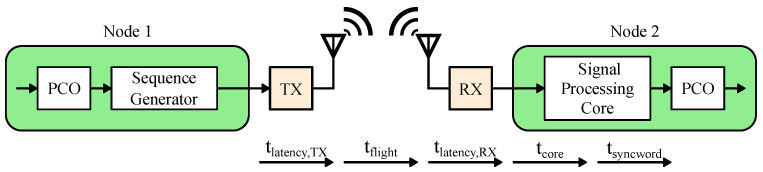
Visual illustration of Equation (Equation 1). The latencies are due to the following: receive and transmit digital filter delays, speed of light propagation delay, and the syncword duration.

**Figure 13 sensors-22-05324-f013:**
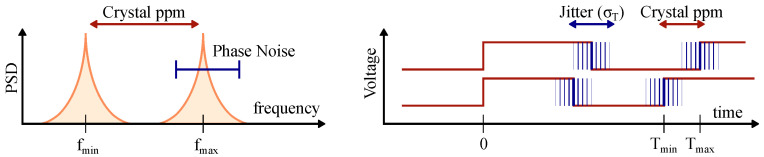
Visual representation of the reference crystal non-idealities. Phase noise in frequency domain is equivalent to jitter in the time domain.

**Figure 14 sensors-22-05324-f014:**
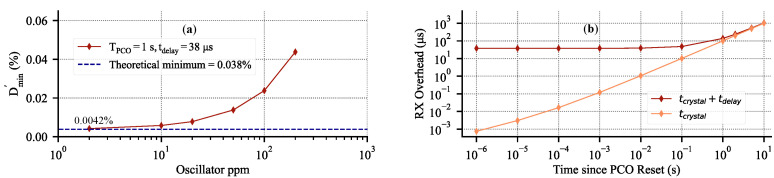
Simulated duty-cycle limits in a 50-node network randomly distributed on a 100 m square grid. (**a**) $D_{min}^{\prime }$ vs. oscillator stability; values approach theoretical minimum for a near-perfect crystal. (**b**) RX window size overhead to receive the syncword assuming 50 ppm crystal with RMS jitter of 25 ps. Notably after 1 s, $t_{delay}$ becomes negligible and crystal non-ideality limits the minimum duty-cycle.

**Figure 16 sensors-22-05324-f016:**
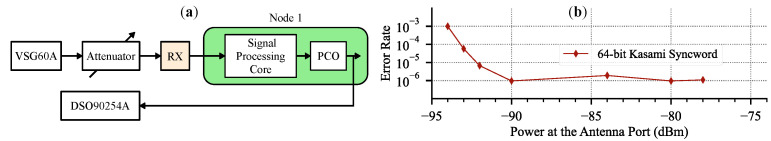
(**a**) Syncword error rate measurement setup. (**b**) Syncword detection error rate vs. signal strength at the antenna port.

**Figure 18 sensors-22-05324-f018:**
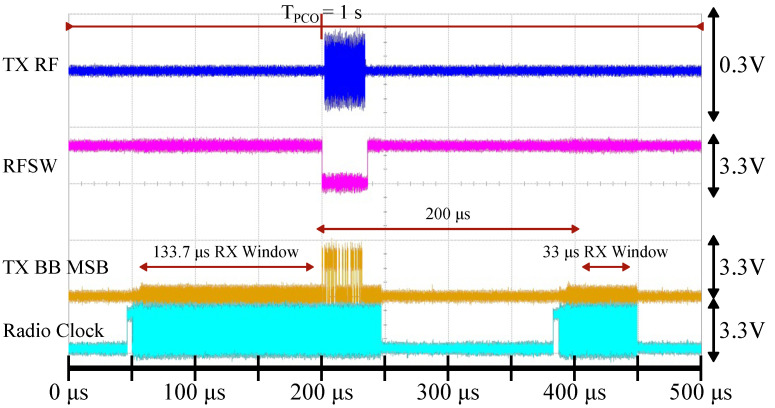
Synchronized syncword detection and duty-cycled communication demonstration. A total of 200 μs after the PCO is reset, the radio wakes up again to open the RX window for 33 μs. This is sufficient time to detect a 32 μs sequence because timing errors have just been reset and uncertainty due to $t_{crystal}$ has not yet accumulated. Signals from top to bottom: TX RF output at the antenna (blue), RF switch direction where logic high selects RX (purple), TX baseband data (MSB) output of the chip (yellow), AD9364 clock (teal).

**Figure 19 sensors-22-05324-f019:**
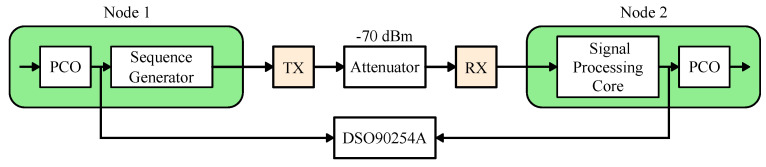
Measurement setup to quantify the limits of duty-cycling (Figure 20).

**Figure 21 sensors-22-05324-f021:**
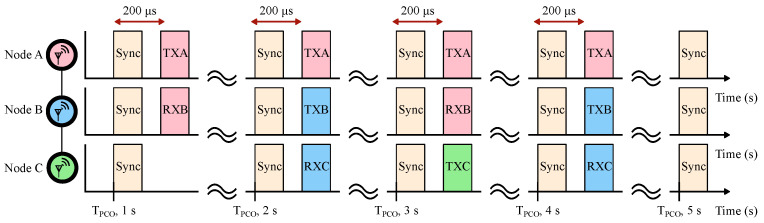
A three-node multi-hop measurement setup. Upon reaching synchronization, node A starts constantly broadcasting a TXA data packet. If node B receives the TXA, it will transmit its own TXB packet on the following PCO cycle. If node C receives the TXB, it will generate a TXC packet. Generation of the TXC packet indicates a successful multi-hop data transfer.

**Table 1 sensors-22-05324-t001:** Selected list of the current BLE mesh solutions.

**Protocol**	Trickle [18]	FruityMesh [19]	BLEMesh [20]	MRT-BLE [21]	BOM [22]
**Type**	Flooding	Connection	Opportunistic flooding	Connection	Flooding
**Protocol**	BLUES [23]	BMADS [24]	BLE-EEOR [25]	NUPFA [26]	BLE Mesh [4]
**Type**	Connection	Managed flooding	Opportunistic flooding	Connection	Managed flooding

**Table 2 sensors-22-05324-t002:** Design trade-off space. Low crystal stability implies wide frequency variation of individual nodes, which enhances synchronization speed and stability (leader node is faster than second-fastest node) at the cost of increased duty-cycle. High syncword duration implies higher SNR and longer synchronization range at the cost of higher power. Arrows (↑↓) show relationships; colors indicate ↑ neutral, ↑ wanted, ↑↓ unwanted.

**Crystal Stability:**	↑ ppm	↑ Sync Speed	↑ Sync Stability	↑ $D_{min}$
**Syncword:**	↑ length	↓ Sync Speed	↑ Network Range	↑ $D_{min}$

**Table 3 sensors-22-05324-t003:** Per-component power breakdown for continuous vs. duty-cycled operation at $T_{PCO}$ = 1 s corresponding to the measurement in Figure 17. ∗ PCO power decreases slightly with higher count due to decreased switching. † TX and RX fully-on in FDD mode. ‡ Power is dominated by leakage.

Component	2 MSPS Fully On (μW)	2 MSPS 1 s Duty-Cycled (μW)
19.2 MHz Oscillator	450	450
PCO + Duty Controller	14.7	12.9 ∗
Sequence Generator	1.28	≈0
Signal Processing Core	504	0.347 ‡
AD9364 TX@-10 dBm	427,450 †	517
RF Switch	1.8	1.8
**Total Power**	428,422 †	982
**$D_{min}$ (%)**	100	0.0204

## Data Availability

Not applicable.

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
