# Peer review of "PCO-Based BLE Mesh Accelerator"

_sensors, 2022, doi:10.3390/s22145324_

Round 1
Reviewer 1 Report
1.In the paper a lightweight physical (PHY) layer accelerator to the stack in Bluetooth Low Energy (BLE) Mesh Networks is suggested. The PHY layer accelerator enables scalable synchronization command for multihop mesh networking.
2.The authors analyzed how synchronized BLE nodes can be aggressively duty-cycled to save power even with an over-provisioned AD9364 transceiver.
3.The authors examined the current BLE Mesh solutions and possible challenges are described.
4. The issues and problems of the network synchronization analysis and the requirements for mesh networking are studied.
5. The authors validated the analytical results with the proposed ASIC accelerator.
6.The authors verified a representative mesh configuration to demonstrate a synchronized multi-hop data transmission.
7.Generally, the paper can be accepted. It may be interesting to present the current BLE Mesh solutions (section 2) in a table. This will be useful for readers.
Reviewer 2 Report
This paper proposed a PCO (Pulse-Coupled Oscillator)-based accelerator to augment the BLE (Bluetooth Low Energy) physical layer with a scalable synchronization command to enable multi-hop mesh networking. The PCO-based accelerator is an interesting idea for BLE mesh networks. The following problems need to be addressed to improve the quality of the paper.
(1) The reviewer wonders how to promote the proposed PCO-based accelerator into existing BLE devices of mesh networks. After all, the proposed PCO-based accelerator is in physical layer. Please discuss it.
(2) In Figure 8, the reviewer finds “a) Network connectivity Map”, “b) Network Strobe”, ” (a) Network of 50 …”, and “(b) PCO reset…”. Numbering should be consistent in style.
Reviewer 3 Report
A well written paper with solid formulation and results. Some minor considerations prior publication:
1) Litterature review should be clearly updated, as there are very few cited works that have been published in the last three years.
2) I do not understand the scope of Fig. 13. Is it based on real measurements? Otherwise it can be removed.
3) Which communication protocol do you use for your simulation studies?
